# The variable selection of two-part regression model for semicontinuous data

**Yahui Lu**[1], **Aiyi Liu**[2], **Tao Jiang**[3]*

**1** School of Economics and Management, Zhejiang University of Science and Technology, Hangzhou, Zhejiang, China, **2** Biostatistics and Bioinformatics Branch, *Eunice Kennedy Shriver* National Institute of Child Health and Human Development, Bethesda, Maryland, United States of America, **3** School of Statistics and Mathematics, Zhejiang Gongshang University, Hangzhou, Zhejiang, China

* jtao@263.net

## Abstract

In many research fields, measurement data containing too many zeros are often called semicontinuous data. For semicontinuous data, the most common method is the two-part model, which establishes the corresponding regression model for both the zero-valued part and the nonzero-valued part. Considering that each part of the two-part regression model often encounters a large number of candidate variables, the variable selection becomes an important problem in semicontinuous data analysis. However, there is little research literature on this topic. To bridge this gap, we propose a new type of variable selection methods for the two-part regression model. In this paper, the Bernoulli-Normal two-part (BNT) regression model is presented, and a variable selection method based on Lasso penalty function is proposed. To solve the problem that Lasso estimator does not have Oracle attribute, we then propose a variable selection method based on adaptive Lasso penalty function. The simulation results show that both methods can select variables for BNT regression model and are easy to implement, and the performance of adaptive Lasso method is superior to the Lasso method. We demonstrate the effectiveness of the proposed tools using dietary intake data to further analyze the important factors affecting dietary intake of patients.

## 1 Introduction

In many research fields, measurement data containing too many zeros are often called semicontinuous data. Aitchison (1955) [1] pointed out that semicontinuous data can be regarded as generated by a mixed distribution consisting of a certain proportion of the zero-valued data (degenerate distribution) and the nonzero continuous data (continuous distribution). For instance, in the study of dietary intake of patients, some food components are consumed almost daily by patients, while others are consumed occasionally, resulting in many zeros in the intake data (Lu et al., 2020 [2]). In nonlife insurance, claim outcomes generally contain a probability mass at zero, indicating no claim occurrence (Frees et al., 2013 [3]; Yang,

**Data availability statement:** Data cannot be shared publicly because of confidentiality of patients. Data are available from the Eunice Kennedy Shriver National Institutional Data

Access/Ethics Committee (contact via nanselt@mail.nih.gov) for researchers who meet the criteria for access to confidential data.

**Funding:** This work is supported by Zhejiang University of Science and Technology Special Fund for Basic Scientific Research [No. 2025QN084] awarded to Y.L., the Research Project of Zhejiang Federation of Humanities and Social Sciences [No. 2025N072] awarded to Y.L., and in part by the Hangzhou Philosophy and Social Science Planning Project [No. Z23JC042]. The funders all play an important role in the study research design, data collection and analysis.

**Competing interests:** The authors have declared that no competing interests exist.

2022 [4]). In addition, examples of semicontinuous outcomes include health care expenditures with zero representing no utilization (Smith et al., 2017 [5]; Huling et al., 2021 [6]), rainfall amounts with zero representing no rain (Hyndman and Grunwald, 2000 [7]), and alcohol consumption (Liu et al., 2008 [8]), among many others.

For semicontinuous data, too many zero values lead to severe right-biased data distribution, making it difficult to fit the data by the traditional distribution model. In this case, there are Tobit model, sample selection model and two-part model available for semicontinuous data. Among these models, the two-part model is the most commonly used method, which regards data as generated by two different random processes. The first process is usually considered as the binary part of the data, which considers whether the zero value occurs, indicating whether a certain behavior has occurred, and this process can be assumed to follow Bernoulli distribution. The second process is often referred to as the continuous part of the data, which considers the generation of nonzero values, and this process can be assumed to follow some kind of continuous distribution, such as normal distribution, gamma distribution, etc. Specifically, let $X = \{X_1, X_2, \ldots, X_n\}$ follow a semicontinuous distribution with probability $\pi$ of being nonzero and density function $g(X \mid X > 0)$ for the positive part, then the two-part model is constructed as

$$f(x_i) = (1 - \pi)I(x_i = 0) + [\pi g(x_i \mid x_i > 0)]I(x_i > 0) \tag{1}$$
$$x_i \geq 0, i = 1, 2, \ldots, n,$$

where $0 \leq \pi \leq 1$, $I(\cdot)$ denotes the indicator function. In Eq (1), when $\pi = 0$, all data is zero, and all data is nonzero which can be assumed to follow a continuous distribution when $\pi = 1$. In general, we assume that $\pi$ is strictly between 0 and 1 to ensure the data set contains a certain number of nonzero values.

To understand the relationship between a semicontinuous outcome and a set of predictors, two-part regression models are typically used, which model the zero part and positive part separately. In general, the probability mass at zero can be characterized by a logistic regression, and the positive part can be modeled using distributions of positive random variables such as lognormal, gamma, and generalized beta of the second kind (GB2) distributions. Since each part may encounter different and a large number of candidate variables, the problem of variable selection may arise in the two-part regression models of semicontinuous data. To date, researchers have proposed a number of methods to select variables for zero-inflated count data which are similar to semicontinuous data (Zeng et al., 2014 [9]; Wang et al., 2015 [10]; Cantoni and Auda, 2018 [11]; Lee et al., 2020 [12]), but there is little literature on variable selection of semicontinuous data. Han et al. (2018) [13] sought a feasible way of conducting variable selection for random effects two-part models. But the models raise computational challenges in fitting due to numerical integration of the related random effects. Feng and Boyle (2021) [14] proposed a sparse group lasso regularization method for the selection of groups of variables in two-part models. Because they focused on whether a group of variables contribute to the whole model, the sparse group lasso regularization was combined with the two-part model, where the group of variables were carefully constructed to deal with the underlying relationships corresponding to group coefficients. However, this method only allows a group of variables to be selected or excluded, and cannot select the important variables within the group. Therefore, this approach is not ideal in some applications. For example, when studying factors that influence the occurrence of a disease, a gene is described by a group of variables, but it is clear that not every variable has a significant effect. By selecting genes (variables) to extract those genes that directly affect the classification accuracy and

ignoring those genes that have no impact on the classification accuracy, computational performance and classification accuracy can be significantly improved. In addition, because the sub-models in the two-part models may contain different variables, or there is no obvious group structure between variables, the proposed method may not have good variable selection effect. In the two sub-models of two-part models, we only focus on whether a single variable has an effect, and propose two methods based on penalty functions to select variables for semicontinuous data. Our proposed methods are easy to implement and effectively select the important variables.

The rest of the article is arranged as follows. In Sect 2, we introduce the Bernoulli-Normal regression model for semicontinuous data, and a Gauss-Newton iteration method is given to estimate parameters. In Sect 3, we propose a new type of variable selection methods of Lasso, adaptive Lasso penalizing the Bernoulli-Normal regression model respectively, and a coordinate descent algorithm is proposed to estimate the parameters. In Sect 4, we conduct simulation studies to evaluate the performance of the proposed methods. Real data from a dietary intervention trial is used to illustrate the methods in Sect 5, and some concluding remarks are given in Sect 6. All simulations and real data analysis results are conducted by R software, and all codes have been provided in the Appendix (See S1 Appendix).

## 2 Bernoulli-Normal regression model

Consider a set of independent identically distributed samples $\{X_1, X_2, \ldots, X_n\}$ from a semicontinuous population $X$, where $n$ is the sample size. Denote $Y_i = I(X_i > 0)$, $Y = \{Y_1, Y_2, \ldots, Y_n\}$, $i = 1, 2, \ldots, n$, where $I(\cdot)$ is the indicator function. Based on the basic idea of constructing a two-part model, we can divide the model into two parts. For the first part, whether $X$ is zero can be treated as being from a Bernoulli distribution, that is, it is assumed that $Y$ follows a Bernoulli distribution; For the second part, the nonzero part is assumed to follow a normal distribution. In practical application, considering the nonzero part has a certain skewness, then a logarithmic transformation for $X > 0$ is generally applied. According to the above construction process, the Bernoulli-Normal two-part (BNT) model is established as

$$f(x_i) = (1 - \pi)^{1-y_i} \times \left[ \pi N(x_i; \mu, \sigma^2) \right]^{y_i}, \tag{2}$$
$$x_i \geq 0, 0 \leq \pi \leq 1,$$

where $\pi = \Pr(X > 0)$ represents the mixing ratio, that is, the proportion of nonzero continuous data; $N(X; \mu, \sigma^2)$ is a normal density function with mean $\mu$ and variance $\sigma^2$. According to the skewness of specific data, other distributions can also be considered to replace the normal distribution in Eq (2), such as gamma distribution, skewed normal distribution, etc. In this paper, we mainly focus on the BNT model.

In BNT model (See Eq (2)), in order to understand the relationship between a semicontinuous outcome and a set of predictors, the BNT regression model is established as

$$f(x_i) = (1 - \pi_i)^{1-y_i} \left[ \pi_i N(x_i; \mu_i, \sigma^2) \right]^{y_i}, \tag{3}$$
$$x_i \geq 0, 0 \leq \pi_i \leq 1,$$

$$\mathrm{logit}(\pi_i) = \mathrm{logit}\left[ \Pr(X_i > 0) \right] = z_{1i}^T \beta_1, \tag{4}$$
$$i = 1, 2, \ldots, n.$$

$$\mu_i = E[X_i | X_i > 0] = z_{2i}^T \beta_2, \tag{5}$$

$$i = 1, 2, \dots, n.$$

Where $z_{1i} = (z_{1i0}, z_{1i1}, \dots, z_{1iq_1})^T$ is a $q_1 + 1$ dimensional covariable vector of the mixing proportion $\pi_i$, and $\beta_1 = (\beta_{10}, \beta_{11}, \dots, \beta_{1q_1})^T$ is the corresponding $q_1 + 1$ dimensional coefficient vector. Similarly, $z_{2i} = (z_{2i0}, z_{2i1}, \dots, z_{2iq_2})^T$ is a $q_2 + 1$ dimensional covariable vector of the mean parameter $\mu_i$, and $\beta_2 = (\beta_{20}, \beta_{21}, \dots, \beta_{2q_2})^T$ is the corresponding $q_2 + 1$ dimensional coefficient vector. Set $z_{1i0} = z_{2i0} = 1$ in Eqs (4) and (5), then $\beta_{10}$ and $\beta_{20}$ represent the intercept terms of the two sub-regression parts respectively. In addition, the covariable vector $z_{1i}$ and $z_{2i}$ can be same or different in instance data.

## 2.1 Likelihood function of the BNT regression model

Based on Eqs (4) and (5), we get the likelihood function of the BNT regression model

$$L(\pi_i, \mu_i | x) = \prod_{i=1}^n (1 - \pi_i)^{1 - y_i} \left\{ \frac{\pi_i}{\sqrt{2\pi}\sigma} \exp\left[ -\frac{1}{2\sigma^2}(x_i - \mu_i)^2 \right] \right\}^{y_i}, \tag{6}$$

where $y_i = I(x_i > 0)$, $I(\cdot)$ is the indicator function.

Note that

$$\mu_i = z_{2i}^T \beta_2,$$

$$\pi_i = \frac{\exp(z_{1i}^T \beta_1)}{1 + \exp(z_{1i}^T \beta_1)},$$

it's easy to derive

$$1 - \pi_i = \frac{1}{1 + \exp(z_{1i}^T \beta_1)}, \tag{7}$$

$$\ln \pi_i = z_{1i}^T \beta_1 - \ln[1 + \exp(z_{1i}^T \beta_1)], \tag{8}$$

$$\ln(1 - \pi_i) = -\ln[1 + \exp(z_{1i}^T \beta_1)]. \tag{9}$$

By substituting Eqs (7)–(9) into Eq (6), the log-likelihood function of the BNT regression model is obtained as

$$
\begin{aligned}
l(\theta | x) &= \sum_{i=1}^n \left\{ y_i z_{1i}^T \beta_1 - \ln[1 + \exp(z_{1i}^T \beta_1)] \right\} \\
&\quad + \sum_{i=1}^n \left\{ I(y_i = 1) \left[ -\frac{1}{2\sigma^2}(x_i - z_{2i}^T \beta_2)^2 - \ln \sigma - \frac{1}{2} \ln(2\pi) \right] \right\} \\
&= l_1(\beta_1) + l_2(\beta_2, \sigma),
\end{aligned} \tag{10}
$$

where

$$\theta = (\beta_1^T, \beta_2^T, \sigma)^T,$$

$$l_1(\beta_1) = \sum_{i=1}^{n} \{y_i z_{1i}^T \beta_1 - \ln[1 + \exp(z_{1i}^T \beta_1)]\},$$

$$l_2(\beta_2, \sigma) = \sum_{i=1}^{n} \{I(y_i = 1)[-\frac{1}{2\sigma^2}(x_i - z_{2i}^T \beta_2)^2 - \ln \sigma - \frac{1}{2}\ln(2\pi)]\}.$$

In Eq (10), it is obvious that $l(\theta|\mathbf{x})$ is divided into two independent parts. The first part $l_1(\beta_1)$ is the binomial part, corresponding to the log-likelihood function of the logistic regression, which can be used to estimate the parameter $\beta_1$. The second part $l_2(\beta_2, \sigma)$ is the continuous part, corresponding to the log-likelihood function of the general linear regression, which can be used to estimate the parameters $\beta_2$ and $\sigma$. In this case, the maximization of the log-likelihood function is equivalent to maximize $l_1(\beta_1)$ and $l_2(\beta_2, \sigma)$ respectively, that is,

$$\hat{\beta}_1 = \arg\max_{\beta_1} l_1(\beta_1),$$

$$(\hat{\beta}_2, \hat{\sigma}) = \arg\max_{\beta_2, \sigma} l_2(\beta_2, \sigma).$$

## 2.2 Gauss-Newton iterative parameter estimation method

At present, there are many parameter estimation methods for two-part regression model. In practical application, the specific parameter estimation method is determined by the purpose of investigation and the form selected by each part, among which the maximum likelihood method is one of the most commonly used tools, and its basic algorithm is Gauss-Newton iteration method. To this end, the Gauss-Newton iterative estimation process of BNT regression model is given below. In Eq (10), since $l(\theta|\mathbf{x})$ is divided into two independent parts, we use the Gauss-Newton iteration method to estimate the parameters in $l_1(\beta_1)$ and $l_2(\beta_2, \sigma)$ respectively.

Define the score function of parameter $\beta_1$ as

$$\mathbf{U}(\beta_1) = \frac{\partial l_1(\beta_1)}{\partial \beta_1}.$$

With

$$l_1(\beta_1) = \sum_{i=1}^{n} \{y_i z_{1i}^T \beta_1 - \ln[1 + \exp(z_{1i}^T \beta_1)]\},$$

we get

$$\mathbf{U}(\beta_1) = \sum_{i=1}^{n} \left\{ -\frac{1}{1 + \exp(z_{1i}^T \beta_1)} + y_i \right\} z_{1i}. \tag{11}$$

Define the observation information matrix of parameter $\beta_1$ as

$$\mathbf{I}(\beta_1) = -\frac{\partial^2 l_1(\beta_1)}{\partial \beta_1 \partial \beta_1^T}.$$

It follows that

$$I(\beta_1) = \sum_{i=1}^{n} \{-\frac{1}{[1 + \exp(z_{1i}^T \beta_1)]^2}\} z_{1i} z_{1i}^T. \tag{12}$$

Therefore, based on Eqs (11) and (12), the maximum likelihood estimate $\hat{\beta}_1$ of parameter $\beta_1$ can be obtained through the following iterative equation

$$\hat{\beta}_1^{(t+1)} = \hat{\beta}_1^{(t)} + \boldsymbol{\Gamma}^{-1}(\hat{\beta}_1^{(t)}) U(\hat{\beta}_1^{(t)}), \tag{13}$$

where $\hat{\beta}_1^{(t)}$, $t = 1, 2, \ldots$ represents the parameter iteration value obtained at the $t$th step.

Define the score function of parameter $\omega = (\beta_2^T, \sigma)^T$ as

$$U(\omega) = (U_{\beta_2}^T, U_\sigma)^T = \frac{\partial l_2(\omega)}{\partial \omega}.$$

Since

$$l_2(\omega) = \sum_{i=1}^{n} \{I(y_i = 1)[-\frac{1}{2\sigma^2}(x_i - z_{2i}^T \beta_2)^2 - \ln\sigma - \frac{1}{2}\ln(2\pi)]\},$$

we get

$$U_{\beta_2} = \sum_{i=1}^{n} \{I(y_i = 1)\frac{1}{\sigma^2}(x_i - z_{2i}^T \beta_2) z_{2i}\}, \tag{14}$$

$$U_\sigma = \sum_{i=1}^{n} \{I(y_i = 1)\frac{1}{\sigma^3}[(x_i - z_{2i}^T \beta_2)^2 - \sigma^2]\}. \tag{15}$$

Define the observation information matrix of parameter $\omega = (\beta_2^T, \sigma)^T$ as

$$I(\omega) = -\frac{\partial^2 l_1(\omega)}{\partial \omega \partial \omega^T}.$$

We have

$$I(\omega) = \begin{bmatrix} I_{\beta_2 \beta_2} & I_{\beta_2 \sigma} \\ I_{\beta_2 \sigma}^T & I_{\sigma\sigma} \end{bmatrix}, \tag{16}$$

where

$$-I_{\beta_2 \beta_2} = \sum_{i=1}^{n} \{I(y_i = 1)\frac{1}{\sigma^2} z_{2i} z_{2i}^T\},$$

$$-I_{\beta_2 \sigma} = \sum_{i=1}^{n} \{-I(y_i = 1)\frac{1}{2\sigma^3}(x_i - z_{2i}^T \beta_2)\} z_{2i},$$

$$-I_{\sigma\sigma} = \sum_{i=1}^{n} \{I(y_i = 1)[-\frac{3}{\sigma^4}(x_i - z_{2i}^T \beta_2)^2 + \frac{1}{\sigma^2}]\}.$$

Similarly, based on Eqs (14)–(16), the maximum likelihood estimate value $\hat{\omega} = (\hat{\beta}_2^T, \hat{\sigma})^T$ of parameter $\omega = (\beta_2^T, \sigma)^T$ can be obtained through the following iterative equation

$$\hat{\beta}_1^{(t+1)} = \hat{\beta}_1^{(t)} + \boldsymbol{\Gamma}^{-1}(\hat{\beta}_1^{(t)}) U(\hat{\beta}_1^{(t)}), \tag{17}$$

where $\hat{\omega}^{(t)}$, $t = 1, 2, \ldots$ represents the parameter iteration value obtained at the $t$th step. In addition, it should be noted that the observation information matrix can also be replaced by Fisher information matrix in Eqs (13) and (17).

## 3 Variable selection of BNT regression model

At present, the methods based on penalty function are widely used in variable selection problems. Considering the advantages of Lasso and adaptive Lasso penalty function, the following variable selection methods for BNT regression model are proposed.

### 3.1 Lasso and adaptive Lasso penalized likelihood function

Fan and Li (2001) [15] adopted the penalized likelihood function method for variable selection, and showed that the logarithmic loss function plus penalty function is the most effective variable selection method, which is also called penalized likelihood function method. By selecting an appropriate model, the penalized likelihood function method generally has the following form

$$\min_{\beta} -l(\beta) + \sum_{j=1}^{p} P_\lambda(|\beta_j|), \tag{18}$$

where $l(\cdot)$ is the likelihood function, and it is generally taken as a log-likelihood function, as is the case in this paper. $P_\lambda(|\beta_j|)$ is the penalty function for $\lambda$, and $\lambda > 0$ is the tuning parameter represents the magnitude of penalty.

In the regression analysis, it is found that the estimator obtained by the ordinary least square method usually has a small deviation and a large variance, so the final prediction accuracy is not ideal. In order to improve the accuracy of prediction, researchers suggest that some regression coefficients can be compressed or set to 0. The basic idea is to improve the accuracy of prediction by sacrificing part of the bias. Therefore, Tibshirani (1996) [16] proposed the Lasso penalty function

$$P_\lambda(|\beta_j|) = \lambda|\beta_j|,$$

where $\lambda > 0$ is tuning parameter.

According to the no-penalty log-likelihood function (See Eq (10)), the Lasso penalized log-likelihood function of BNT regression model is obtained as

$$l_{lasso}(\beta_1, \beta_2, \sigma; \lambda_1, \lambda_2) = l(\beta_1, \beta_2, \sigma) - \lambda_1 \sum_{j=1}^{q_1} |\beta_{1j}| - \lambda_2 \sum_{s=1}^{q_2} |\beta_{2s}|$$

$$= l_{1,lasso}(\beta_1, \lambda_1) + l_{2,lasso}(\beta_2, \lambda_2), \tag{19}$$

where

$$l_{1,lasso}(\beta_1, \lambda_1) = \sum_{i=1}^{n}\{y_i z_{1i}^T \beta_1 - \ln[1 + \exp(z_{1i}^T \beta_1)]\} - \lambda_1 \sum_{j=1}^{q_1} |\beta_{1j}|,$$

$$l_{2,lasso}(\beta_2, \lambda_2) = \sum_{i=1}^{n}\{I(y_i = 1)[-\frac{1}{2\sigma^2}(x_i - z_{2i}^T \beta_2)^2 - \ln\sigma - \frac{1}{2}\ln(2\pi)]\}$$

$$- \lambda_2 \sum_{s=1}^{q_2} |\beta_{2s}|,$$

$\lambda_1 \geq 0$ and $\lambda_2 \geq 0$ are tuning parameters. Besides, Eq (19) does not penalize intercept parameters $\beta_{10}$, $\beta_{20}$ and variance parameter $\sigma$.

In Eq (19), the penalized log-likelihood function of BNT regression model can be divided into two independent parts $l_{1,lasso}(\beta_1, \lambda_1)$ and $l_{2,lasso}(\beta_2, \lambda_2)$, where $l_{1,lasso}(\beta_1, \lambda_1)$ is regarded as the Lasso penalized log-likelihood function of binary part logistic regression, and $l_{2,lasso}(\beta_2, \lambda_2)$ is regarded as the Lasso penalized log-likelihood function of continuous part normal regression. Since $l_{1,lasso}(\beta_1, \lambda_1)$ and $l_{2,lasso}(\beta_2, \lambda_2)$ respectively contain parameters $\beta_1$ and $\beta_2$, the optimal estimator of Eq (19) can be respectively obtained by

$$\hat{\beta}_{1,lasso}(\lambda_1) = \arg\max_{\beta_1} l_{1,lasso}(\beta_1, \lambda_1),$$

$$\hat{\beta}_{2,lasso}(\lambda_2) = \arg\max_{\beta_2} l_{2,lasso}(\beta_2, \lambda_2).$$

Although the method based on Lasso penalty function can efficiently select variables, its estimators are biased and do not satisfy the Oracle properties. Therefore, Zou (2006) [17] proposed the adaptive Lasso penalty function

$$P_\lambda(|\beta_j|) = \lambda \varpi_j |\beta_j|,$$

where $\varpi_j = 1/|\hat{\beta}_j|^\gamma$ is penalty weight, $\hat{\beta}_j$ is a consistent estimate of the parameter $\beta_j$ obtained without penalty, and generally set $\gamma = 1$. The adaptive Lasso penalty function applies different penalty weights to the coefficients of different covariables. In addition, Zou (2006) [17] has proved that the adaptive Lasso estimators have Oracle properties.

According to the no-penalty log-likelihood function (See Eq (10)), the adaptive Lasso penalized log-likelihood function of BNT regression model is obtained as

$$l_{alasso}(\beta_1, \beta_2, \sigma; \lambda_1, \lambda_2) = l(\beta_1, \beta_2, \sigma) - \lambda_1 \sum_{j=1}^{q_1} \omega_{1j}|\beta_{1j}| - \lambda_2 \sum_{s=1}^{q_2} \omega_{2s}|\beta_{2s}|$$

$$= l_{1,alasso}(\beta_1, \lambda_1) + l_{2,alasso}(\beta_2, \lambda_2), \tag{20}$$

where

$$l_{1,alasso}(\beta_1, \lambda_1) = \sum_{i=1}^{n}\{y_i z_{1i}^T \beta_1 - \ln[1 + \exp(z_{1i}^T \beta_1)]\} - \lambda_1 \sum_{j=1}^{q_1} \omega_{1j}|\beta_{1j}|,$$

$$l_{2,alasso}(\beta_2, \lambda_2) = \sum_{i=1}^{n}\{I(y_i = 1)[-\frac{1}{2\sigma^2}(x_i - z_{2i}^T \beta_2)^2 - \ln\sigma - \frac{1}{2}\ln(2\pi)]\}$$

$$- \lambda_2 \sum_{s=1}^{q_2} \omega_{2s}|\beta_{2s}|,$$

and $\omega = (\omega_1^T, \omega_2^T)^T$, $\omega_1 = (\omega_{11}, \ldots, \omega_{1q_1})^T$, $\omega_2 = (\omega_{21}, \ldots, \omega_{2q_2})^T$ are the weight coefficients. $\lambda_1 \geq 0$, $\lambda_2 \geq 0$ are tuning parameters. Similarly, Eq (20) does not penalize intercept parameters $\beta_{10}$, $\beta_{20}$ and variance parameter $\sigma$. In addition, we set the weight coefficient as $\omega = 1/|\hat{\theta}_{ml}|$, where $\hat{\theta}_{ml}$ is the maximum likelihood estimate of the parameter $\theta$. And it should be noted that Eq (20) is reduces to the Lasso penalized log-likelihood function of BNT regression

model (19) when $\omega = 1$. Therefore, the Lasso method can be regarded as a special case of the adaptive Lasso method.

Similar to the previous discussion, the adaptive Lasso penalized log-likelihood function of BNT regression model (20) can be divided into two independent parts $l_{1,alasso}(\beta_1, \lambda_1)$ and $l_{2,alasso}(\beta_2, \lambda_2)$, where $l_{1,alasso}(\beta_1, \lambda_1)$ is regarded as the adaptive Lasso penalized log-likelihood function of the binary part logistic regression, and $l_{2,alasso}(\beta_2, \lambda_2)$ is regarded as the adaptive Lasso penalized log-likelihood function of continuous part normal regression. Since $l_{1,alasso}(\beta_1, \lambda_1)$ and $l_{2,alasso}(\beta_2, \lambda_2)$ respectively contain parameters $\beta_1$ and $\beta_2$, the optimal estimator of Eq (20) can be respectively obtained by

$$\hat{\beta}_{1,alasso}(\lambda_1) = \underset{\beta_1}{\arg\max}\, l_{1,alasso}(\beta_1, \lambda_1),$$

$$\hat{\beta}_{2,alasso}(\lambda_2) = \underset{\beta_2}{\arg\max}\, l_{2,alasso}(\beta_2, \lambda_2).$$

As mentioned above, the Lasso penalized log-likelihood function can be regarded as a special case of the adaptive Lasso penalized log-likelihood function. Therefore, below we only introduce the parameters estimation procedure of the adaptive Lasso penalty likelihood function.

## 3.2 The coordinate descent method for parameters estimation

At present, many scholars have proposed efficient estimation algorithms for the Lasso method and adaptive Lasso method, such as least angle regression algorithm (Efron et al., 2004 [18]; Friedman et al., 2007 [19]), coordinate descent algorithm (Wu and Lange, 2008 [20]; Friedman et al., 2010 [21]), etc. In this paper, we use the coordinate descent method to optimize the estimation of the adaptive Lasso penalized log-likelihood function (20). In addition, since $l_{2,alasso}(\beta_2, \lambda_2)$ is the simplest linear regression and $l_{1,alasso}(\beta_1, \lambda_1)$ in the generalized linear regression, the optimal solution process of $l_{2,alasso}(\beta_2, \lambda_2)$ is given first, and then the optimal solution of $l_{1,alasso}(\beta_1, \lambda_1)$ is given.

Based on the log-likelihood function of BNT (10), the normal distribution log-likelihood function of the continuous part without penalty term is

$$l_2(\beta_2) = \sum_{i=1}^{n} \{I(y_i = 1)[-\frac{1}{2\sigma^2}(x_i - z_{2i}^T\beta_2)^2 - \ln\sigma - \frac{1}{2}\ln(2\pi)]\}.$$

By removing the last two terms in $l_2(\beta_2)$ which are not related to $\beta_2$, the maximum likelihood estimate of the parameter $\beta_2$ is equivalent to the least squares estimate of

$$\min_{\beta_2} \sum_{i=1}^{n} \{I(y_i = 1)(x_i - z_{2i}^T\beta_2)^2\}$$

about the parameter $\beta_2$. Therefore, with the addition of the adaptive Lasso penalty function, the optimization $l_{2,alasso}(\beta_2, \lambda_2)$ is equivalent to

$$\min_{\beta_2} R(\beta_2, \lambda_2) = \min_{\beta_2} \sum_{i=1}^{n} \{I(y_i = 1)(x_i - z_{2i}^T\beta_2)^2 + \lambda_2 \sum_{s=1}^{q_2} \omega_{2s}|\beta_{2s}|\}, \tag{21}$$

where

$$R(\beta_2, \lambda_2) = \sum_{i=1}^{n} \{ I(y_i = 1)(x_i - z_{2i}^T \beta_2)^2 + \lambda_2 \sum_{s=1}^{q_2} |\beta_{2s}| \}.$$

In addition, since the coordinate descent method needs to be updated for one variable dimension each time, in order to facilitate the derivation, each variable of parameter $\beta_2$ in $R(\beta_2, \lambda_2)$ is expressed, that is,

$$\begin{aligned} R(\beta_2, \lambda_2) &= \sum_{i=1}^{n} \{ I(y_i = 1)(x_i - z_{2i}^T \beta_2)^2 + \lambda_2 \sum_{s=1}^{q_2} \omega_{2s} |\beta_{2s}| \} \\ &= \sum_{i=1}^{n} \{ I(y_i = 1)(x_i - \beta_{20} - \sum_{s=1}^{q_2} z_{2is} \beta_{2s})^2 + \lambda_2 \sum_{s=1}^{q_2} \omega_{2s} |\beta_{2s}| \}. \end{aligned} \quad (22)$$

Next, the coordinate descent method is used to solve the optimal value of Eq (22). In the $t$th iteration, it is assumed that the $k$th variable is being updated and the estimated value $\tilde{\beta}_{2(-k)} = (\tilde{\beta}_{20}^{(t)}, \tilde{\beta}_{21}^{(t)}, ..., \tilde{\beta}_{2(k-1)}^{(t)}, \tilde{\beta}_{2(k+1)}^{(t-1)}, ..., \tilde{\beta}_{2q_2}^{(t-1)})^T$ has been obtained, where $\tilde{\beta}_{2h}^{(t)}$ represents the iteration value of the $h$th variable ($h \neq k$) in the $t$th iteration.

Firstly, taking the partial derivative of Eq (22) with the parameter $\beta_{2k}$, we get

$$\begin{aligned} \frac{\partial R(\beta_2, \lambda_2)}{\partial \beta_{2k}} \Big|_{\tilde{\beta}_{2(-k)}} = \sum_{i=1}^{n} \{ &-2I(y_i = 1)z_{2ik}(x_i - z_{2i(-k)}^T \tilde{\beta}_{2(-k)} - z_{2ik}\beta_{2k}) \\ &+ \lambda_2 \omega_{2k} \mathrm{sign}(\beta_{2k}) \}, \end{aligned} \quad (23)$$

where $z_{2i(-k)} = (z_{2i0}, z_{2i1}, ..., z_{2i(j-1)}, z_{2i(j+1)}, ..., z_{2iq_2})^T$, and $\mathrm{sign}(\cdot)$ is the sign function.

Then, setting Eq (23) equal to 0 and through a series of deductions, we get the updated value of the $k$th variable in the $t$th iteration

$$\tilde{\beta}_{2k}^{(t)} = \frac{S(2\sum_{i=1}^{n} \{ I(y_i = 1)z_{2ik}(x_i - z_{2i(-k)}^T \tilde{\beta}_{2(-k)}) \}, \lambda_2 \omega_{2k})}{2\sum_{i=1}^{n} I(y_i = 1)z_{2ik}^2}, \quad (24)$$

where $S(a, b)$ is a soft-thresholding operator defined by Donoho and Johnstone (1994) [22], and its concrete form is

$$S(a, b) = \begin{cases} a - b, & a > 0 \; b < |a| \\ a + b, & a < 0 \; b < |a| \\ 0, & b \geq a \end{cases}.$$

In the process of the $t$th iteration, the remaining variables are updated successively according to the update formula (24). Observe the change in each dimension of the $t$th iteration value $\beta_2^{(t)}$ and the $(t-1)$th iteration value $\beta_2^{(t-1)}$. If the change value of all dimensions is small enough or meets certain convergence conditions, then the $t$th iteration value $\beta_2^{(t)}$ is the final parameter estimator. Otherwise, the next iteration is entered according to the update formula (24).

In the following, we use coordinate descent method to optimize $l_{1,alasso}(\beta_1, \lambda_1)$. Based on the log-likelihood function of BNT (10), the logistic regression log-likelihood function of the

binary part without penalty term is

$$l_1(\beta_1) = \sum_{i=1}^{n} \{y_i z_{1i}^T \beta_1 - \ln[1 + \exp(z_{1i}^T \beta_1)]\}.$$

Adding the adaptive Lasso penalty term, the binary part of the adaptive Lasso penalized logistic regression log-likelihood function is obtained as

$$l_{1,alasso}(\beta_1, \lambda_1) = \sum_{i=1}^{n} \{y_i z_{1i}^T \beta_1 - \ln[1 + \exp(z_{1i}^T \beta_1)]\} - \lambda_1 \sum_{j=1}^{q_1} \omega_{1j}|\beta_{1j}|. \qquad (25)$$

In Eq (25), the first term $l_1(\beta_1)$ is a non-convex function and cannot be directly estimated by the coordinate descent method. For the no-penalty logistic regression log-likelihood function $l_1(\beta_1)$, the process of Gauss-Newton method is given in Sect 2. And the iterative formula (17) to solve the optimal parameter is also called iteratively reweighted least squares (IRLS) method. Assuming that the current estimate of parameter $\beta_1$ is $\tilde{\beta}_1^{old}$, the second-order approximation of $l_1(\beta_1)$, namely, the second-order Taylor expansion of $l_1(\beta_1)$ at $\tilde{\beta}_1^{old}$ is obtained as

$$l_Q(\beta_1) = -\sum_{i=1}^{n} w_i(\gamma_i - z_{1i}^T \beta_1)^2 + C^2(\beta_1), \qquad (26)$$

where

$$\gamma_i = z_{1i}^T \tilde{\beta}_1^{old} + \frac{y_i - \tilde{p}^{old}(z_{1i})}{\tilde{p}^{old}(z_{1i})(1 - \tilde{p}^{old}(z_{1i}))},$$

$$w_i = \tilde{p}^{old}(z_{1i})(1 - \tilde{p}^{old}(z_{1i})),$$

$$\tilde{p}^{old}(z_{1i}) = \frac{\exp(z_{1i}^T \tilde{\beta}_1^{old})}{1 + \exp(z_{1i}^T \tilde{\beta}_1^{old})},$$

and $C^2(\beta_1)$ is a constant term. In this case, $\gamma_i$ is called the work response, which changes with each iteration and is a temporary response during the iteration. $w_i$ is called the weighted value which also changes with each iteration.

Since $l_Q(\beta_1)$ is a quadratic convex function and the second order is approximately $l_1(\beta_1)$, the optimal adaptive Lasso penalized logistic regression log-likelihood function (25) is equivalent to minimizing the penalized weighted least squares of the adaptive Lasso, that is,

$$\min R(\beta_1, \lambda_1) = \min \sum_{i=1}^{n} \{w_i(\gamma_i - z_{1i}^T \beta_1)^2 + \lambda_1 \sum_{j=1}^{q_1} \omega_{1j}|\beta_{1j}|\}. \qquad (27)$$

At this time, the optimal solution of Eq (27) can be solved directly by the coordinate descent method, which is similar to the optimal solution of $l_{2,alasso}(\beta_2, \lambda_2)$. In the process of the $t$th iteration, it is assumed that the $k$th variable is being updated and the estimated value is $\tilde{\beta}_{1(-k)} = (\tilde{\beta}_{10}^{(t)}, \tilde{\beta}_{11}^{(t)}, ..., \tilde{\beta}_{1(k-1)}^{(t)}, \tilde{\beta}_{1(k+1)}^{(t-1)}, ..., \tilde{\beta}_{1q_1}^{(t-1)})^T$, where $\tilde{\beta}_{1h}^{(t)}$ represents the iteration value of the $h$th variable ($h \neq k$) in the $t$th iteration.

Firstly, taking the partial derivative of Eq (27) with the parameter $\beta_{1k}$ , we get

$$\frac{\partial R(\beta_1, \lambda_1)}{\partial \beta_{1k}}|_{\tilde{\beta}_{1(-k)}} = \sum_{i=1}^{n}\{-2w_i z_{1ik}(\gamma_i - \boldsymbol{z}_{1i(-k)}^T \tilde{\beta}_{1(-k)} - z_{1ik}\beta_{1k}) + \lambda_1 \omega_{1k}\text{sign}(\beta_{1k})\}, \qquad (28)$$

where $\boldsymbol{z}_{1i(-k)} = (z_{1i0}, z_{1i1}, \ldots, z_{1i(j-1)}, z_{2i(j+1)}, \ldots, z_{1iq_1})^T$, and $\text{sign}(\cdot)$ is the sign function.

Then, setting Eq (28) equal to 0 and through a series of deductions, we get the updated value of the $k$th variable in the $t$th iteration

$$\tilde{\beta}_{1k}^{(t)} = \frac{S(2\sum_{i=1}^{n}\{w_i z_{1ik}(\gamma_i - \boldsymbol{z}_{1i(-k)}^T \tilde{\beta}_{1(-k)})\}, \lambda_1 \omega_{1k})}{2\sum_{i=1}^{n} z_{1ik}^2}, \qquad (29)$$

where $S(a, b)$ is a soft-thresholding operator.

In the process of the $t$th iteration, the remaining variables are updated successively according to the update formula (29). Observe the change in each dimension of the $t$th iteration value $\beta_1^{(t)}$ and the $(t-1)$th iteration value $\beta_1^{(t-1)}$, if the change value of all dimensions is small enough or meets certain convergence conditions, then the $t$th iteration value $\beta_1^{(t)}$ is the final parameter estimator. Otherwise, the next iteration is entered according to the update formula (29).

Combining the second-order Taylor expansion approximation and the coordinate descent method, the optimal solution process of Eq (25) is equivalent to a nested loop sequence. ① External loop: For the current parameter estimated value $\tilde{\beta}_1^{old}$, the second-order Taylor approximation $l_Q(\beta_1)$ is obtained from Eq (26); ② Internal loop: The coordinate descent method is used to estimate the parameters of the penalized weighted least squares (27) of the adaptive Lasso, and the update iteration is carried out according to Eq (29).

## 3.3 Selection of tuning parameter

The Lasso or adaptive Lasso penalized BNT regression model also involves another problem, that is, the selection of tuning parameter $\lambda$. At present, the three commonly used methods are K-fold cross-validation, Generalized cross-validation (GCV) and BIC information criterion. Because $K$-fold cross-validation can effectively avoid overfitting when evaluating model prediction performance, the final results are more convincing, and the calculation process of this method is relatively simple. Therefore, we mainly adopt $K$-fold cross-validation to adjust the choice of parameter $\lambda$, and set $K = 10$ in the simulation studies. The basic idea of this method is to divide data into two parts: ① Training set is used to train model; ② Validation set is used to verify model. This method is very popular in statistical data analysis, and the specific steps are as follows.

(1) The whole sample is divided into $K$ parts equally, denoted as $T_1, T_2, \ldots, T_K$;

(2) Keep $T_1$ as the validation set and the remaining $T_2, \ldots, T_K$ as the training set. Firstly, the model is trained in the training set, then the trained model is fitted in the verification set, and the fitting value of the response variable is denoted as $\hat{x}_i(\boldsymbol{z}_i)$. The estimation error of the model can be expressed as

$$CV_1(\lambda) = |T_1|^{-1} \sum_{x_i \in T_1} (x_i - \hat{x}_i(\boldsymbol{z}_i))^2,$$

where $|T_1|$ is the sample size of $T_1$;

(3) Repeat Step (2), each time retain a set of $T_J, J = 2, \dots, K$ as the verification set, and the remaining as the training set, so a set of $CV_1(\lambda), \dots, CV_K(\lambda)$ is obtained;

(4) Calculate the total mean error of cross validation:

$$CV(\lambda) = \frac{1}{K} \sum_J CV_J(\lambda);$$

(5) The estimate of the tuning parameter $\lambda$ is obtained as

$$\hat{\lambda} = \arg\min_{\lambda > 0} CV_J(\lambda).$$

## 4 Simulations

The section will compare the variable selection effects of the two proposed methods through simulation studies.

### 4.1 Simulation data

Firstly, covariables are generated from the multivariate normal distribution $N_q(0, \Sigma)$, where covariance matrix is $\Sigma$ whose elements are $\rho^{|i-j|}$ $(i, j = 1, \dots, q)$. In the settings below, $\rho = 0, 0.6$, $q = 10, 15, 25$, and the sample size is $n = 150, 300$. Secondly, the model coefficients are set in the simulation studies. The coefficients of some covariables are set to nonzero, which means that the simulation includes the corresponding covariables to generate the response variables. The coefficients of the remaining variables are set to zero, indicating that these variables have no effect on the response variables in the model. Finally, according to the BNT regression model, the real model is assumed as

$$x_i = \text{Binomial}(\pi_i) \times N(\mu_i, 1), \tag{30}$$

where

$$\text{logit}(\pi_i) = z_1^T \beta_1,$$

$$\mu_i = z_2^T \beta_2.$$

we set $z_1 = z_2$ in the following simulation scenarios.

In this section, the following three scenarios are simulated, and each scenario is repeated 500 times (More simulations see S1 Appendix).

(1) When $q = 10$, the regression coefficients in model (30) are set as:

$$
\begin{aligned}
\beta_1 &= (0.2, 0.30, 1, 0.8, 0.4, -0.2, 0, 0, 0, 0, 0)^T, \\
\beta_2 &= (0.1, 1.10, 1, -0.36, 0.6, 0, 0, 0, 0, 0, 0)^T;
\end{aligned}
$$

(2) When $q = 15$, the regression coefficients in model (30) are set as:

$$
\begin{aligned}
\beta_1 &= (0.2, 0.30, 1, 0.8, 0.4, -0.2, 0, 0, 0.6, 0, 0, 0, 0, 0, 0, 0)^T, \\
\beta_2 &= (0.1, 1.10, 1, -0.36, 0.6, 0, 0, 0, 0.8, 0, 0, 0, 0, 0, 0, 0)^T;
\end{aligned}
$$

(3) When $q = 25$, the regression coefficients in model (30) are set as:

$$
\begin{aligned}
\beta_1 \quad &= (0.2, 0.30, 1, 0.8, 0.4, -0.2, 0, 0, 0.6, 0, 0, 0, 0, 0, 0, 0, \\
&\qquad 1.1, -0.2, 0, 0, 0, 0, 0, 0.6, 0, 0)^T, \\
\beta_2 \quad &= (0.1, 1.10, 1, -0.36, 0.6, 0, 0, 0, 0.8, 0, 0, 0, 0, 0, 0, 0, \\
&\qquad 0.9, 0.7, 0, 0, 0, 0, 0, 1.2, 0, 0)^T.
\end{aligned}
$$

## 4.2 Simulation results

The proposed variable selection methods of the BNT regression model are based on the Lasso and adaptive Lasso penalty functions, respectively. The effect of two methods are evaluated by comparing the following 5 statistics, and the logistic regression of the binary part, the normal regression of the continuous part and the whole of two parts are calculated respectively.

(1) Mean square error of prediction (MPSE):

$$
\text{MPSE} = \frac{1}{n} \sum_{i=1}^{n} (x_i - \hat{x}_i)^2,
$$

where $\hat{x}_i$ is the predicted value of response variable $x_i$.

(2) Mean square error of parameter (MSE):

$$
\text{MSE} = \frac{1}{q+1} \sum_{j=0}^{q} (\beta_j - \hat{\beta}_j)^2,
$$

where $\hat{\beta}_j$ is the estimate of the covariable coefficient $\beta_j$.

(3) Define Sensitivity and Specificity as:

$$
\text{Sensitivity}(\beta) = \frac{\sharp\{j : \beta_j \neq 0 \,\&\, \hat{\beta}_j \neq 0\}}{\sharp\{j : \beta_j \neq 0\}},
$$

$$
\text{Specificity}(\beta) = \frac{\sharp\{j : \beta_j = 0 \,\&\, \hat{\beta}_j = 0\}}{\sharp\{j : \beta_j = 0\}},
$$

where $\sharp\{j : A\}$ is the number of $j$ that satisfies the condition $A$.

(4) Combining Sensitivity and Specificity, define Accuracy as:

$$
\text{Accuracy}(\beta) = \frac{\sharp\{j : \beta_j \neq 0 \,\&\, \hat{\beta}_j \neq 0\} + \sharp\{j : \beta_j = 0 \,\&\, \hat{\beta}_j = 0\}}{q},
$$

where $q$ is the number of covariables.

For the 5 evaluation statistics, the MPSE value represents the prediction error of the model, and the smaller the value, the better the model fits. The MSE value indicates the difference between the estimated parameter and the actual parameter, and the smaller the value, the closer the estimated value of the independent variable coefficient is to the actual value. The values of Sensitivity, Specificity and Accuracy are in the interval [0,1] to evaluate the effectiveness of variable selection. The Sensitivity value represents the proportion of selected variables in real important variables. The Specificity value represents the proportion of selected

variables in real unimportant variables. And the Accuracy value represents the proportion of correctly selected and correctly eliminated variables in total variables.

For the three scenario settings, we specifically consider $(\rho = 0, n = 100)$, $(\rho = 0.6, n = 100)$, $(\rho = 0, n = 300)$, $(\rho = 0.6, n = 300)$. The simulation results are shown in Tables 1, 2, 3, 4, 5, and 6. According to the results, it can be concluded

**Table 1. The simulation results of the setting $q = 10, \rho = 0$.**

| Method | MPSE | MSE | Sensitivity | Specificity | Accuracy |
|---|---|---|---|---|---|
| 1. When $n = 150$ | | | | | |
| Binomial Part | | | | | |
| LASSO | 1.1157 | 0.0144 | 0.9940 | 0.4793 | 0.6852 |
| ALASSO | 1.0711 | 0.0112 | 0.9840 | 0.7787 | 0.8608 |
| Normal Part | | | | | |
| LASSO | 1.1434 | 0.0365 | 0.8392 | 0.5808 | 0.7100 |
| ALASSO | 1.1223 | 0.0362 | 0.7520 | 0.7720 | 0.7620 |
| Two Parts | | | | | |
| LASSO | 2.2591 | 0.0254 | 0.9080 | 0.5255 | 0.6976 |
| ALASSO | 2.1934 | 0.0237 | 0.8551 | 0.7756 | 0.8114 |
| 2. When $n = 300$ | | | | | |
| Binomial Part | | | | | |
| LASSO | 1.0593 | 0.0062 | 0.9995 | 0.4697 | 0.6816 |
| ALASSO | 1.0301 | 0.0050 | 0.9975 | 0.7970 | 0.8772 |
| Normal Part | | | | | |
| LASSO | 1.1177 | 0.0186 | 0.9432 | 0.4496 | 0.6964 |
| ALASSO | 1.1017 | 0.0192 | 0.8676 | 0.7628 | 0.8152 |
| Two Parts | | | | | |
| LASSO | 2.1770 | 0.0124 | 0.9682 | 0.4605 | 0.6890 |
| ALASSO | 2.1318 | 0.0121 | 0.9253 | 0.7815 | 0.7718 |

**Table 2. The simulation results of the setting $q = 10, \rho = 0.6$.**

| Method | MPSE | MSE | Sensitivity | Specificity | Accuracy |
|---|---|---|---|---|---|
| 1. When $n = 150$ | | | | | |
| Binomial Part | | | | | |
| LASSO | 1.1163 | 0.0264 | 0.9040 | 0.5390 | 0.6850 |
| ALASSO | 1.0593 | 0.0214 | 0.9235 | 0.7643 | 0.8280 |
| Normal Part | | | | | |
| LASSO | 0.9747 | 0.0492 | 0.7568 | 0.6944 | 0.7256 |
| ALASSO | 0.9581 | 0.0722 | 0.6312 | 0.8000 | 0.7156 |
| Two Parts | | | | | |
| LASSO | 2.0910 | 0.0379 | 0.8222 | 0.6096 | 0.7053 |
| ALASSO | 2.0174 | 0.0468 | 0.7611 | 0.7805 | 0.7718 |
| 2. When $n = 300$ | | | | | |
| Binomial Part | | | | | |
| LASSO | 1.0599 | 0.0129 | 0.9695 | 0.4760 | 0.6734 |
| ALASSO | 1.0403 | 0.0105 | 0.9730 | 0.7617 | 0.8462 |
| Normal Part | | | | | |
| LASSO | 0.9608 | 0.0292 | 0.8232 | 0.6572 | 0.7402 |
| ALASSO | 0.9506 | 0.0379 | 0.7212 | 0.7788 | 0.7500 |
| Two Parts | | | | | |
| LASSO | 2.0207 | 0.0210 | 0.8882 | 0.5584 | 0.7068 |
| ALASSO | 1.9909 | 0.0242 | 0.8331 | 0.7694 | 0.7981 |

**Table 3. The simulation results of the setting $q = 15, \rho = 0$.**

| Method | MPSE | MSE | Sensitivity | Specificity | Accuracy |
|---|---|---|---|---|---|
| 1. When $n = 150$ | | | | | |
| Binomial Part | | | | | |
| LASSO | 1.1735 | 0.0128 | 0.9952 | 0.5128 | 0.6736 |
| ALASSO | 1.0830 | 0.0107 | 0.9828 | 0.7824 | 0.8492 |
| Normal Part | | | | | |
| LASSO | 1.1305 | 0.0338 | 0.8480 | 0.6156 | 0.7085 |
| ALASSO | 1.0955 | 0.0325 | 0.7657 | 0.7991 | 0.7857 |
| Two Parts | | | | | |
| LASSO | 2.3040 | 0.0233 | 0.9149 | 0.5615 | 0.6911 |
| ALASSO | 2.1785 | 0.0216 | 0.8644 | 0.7903 | 0.8175 |
| 2. When $n = 300$ | | | | | |
| Binomial Part | | | | | |
| LASSO | 1.0967 | 0.0057 | 0.9996 | 0.5098 | 0.6731 |
| ALASSO | 1.0369 | 0.0045 | 0.9968 | 0.8182 | 0.8777 |
| Normal Part | | | | | |
| LASSO | 1.0982 | 0.0175 | 0.9320 | 0.5222 | 0.6861 |
| ALASSO | 1.0701 | 0.0157 | 0.8750 | 0.7809 | 0.8185 |
| Two Parts | | | | | |
| LASSO | 2.1948 | 0.0116 | 0.9627 | 0.5157 | 0.6796 |
| ALASSO | 2.1070 | 0.0101 | 0.9304 | 0.8005 | 0.8481 |

**Table 4. The simulation results of the setting $q = 15, \rho = 0.6$.**

| Method | MPSE | MSE | Sensitivity | Specificity | Accuracy |
|---|---|---|---|---|---|
| 1. When $n = 150$ | | | | | |
| Binomial Part | | | | | |
| LASSO | 1.1713 | 0.0241 | 0.9116 | 0.5716 | 0.6849 |
| ALASSO | 1.0926 | 0.0607 | 0.9220 | 0.7646 | 0.8171 |
| Normal Part | | | | | |
| LASSO | 0.9599 | 0.0449 | 0.7640 | 0.7127 | 0.7332 |
| ALASSO | 0.9400 | 0.0646 | 0.6433 | 0.7933 | 0.7333 |
| Two Parts | | | | | |
| LASSO | 2.1312 | 0.0345 | 0.8311 | 0.6384 | 0.7091 |
| ALASSO | 2.0326 | 0.0426 | 0.7700 | 0.7782 | 0.7752 |
| 2. When $n = 300$ | | | | | |
| Binomial Part | | | | | |
| LASSO | 1.0906 | 0.0123 | 0.9644 | 0.5122 | 0.6629 |
| ALASSO | 1.0382 | 0.0088 | 0.9784 | 0.7754 | 0.8431 |
| Normal Part | | | | | |
| LASSO | 0.9359 | 0.0265 | 0.8433 | 0.6469 | 0.7255 |
| ALASSO | 0.9235 | 0.0337 | 0.7517 | 0.7842 | 0.7712 |
| Two Parts | | | | | |
| LASSO | 2.0265 | 0.0194 | 0.8984 | 0.5760 | 0.6942 |
| ALASSO | 1.9617 | 0.0213 | 0.8547 | 0.7796 | 0.8071 |

- From the results of MPSE and MSE, the adaptive Lasso method is slightly better than the Lasso method in the single binary part, the continuous part and the whole of two parts.
- In terms of the results of Sensitivity, both methods have high Sensitivity. As $n$ increases, the Sensitivity of two methods becomes better. From all the results, the Lasso method shows better performance, but when the correlation coefficient $\rho$ is larger, the performance of the adaptive Lasso method becomes better in the binary part.

**Table 5. The simulation results of the setting $q = 25, \rho = 0$.**

| Method | MPSE | MSE | Sensitivity | Specificity | Accuracy |
|---|---|---|---|---|---|
| 1. When $n = 150$ | | | | | |
| Binomial Part | | | | | |
| LASSO | 1.0450 | 0.0389 | 0.8264 | 0.6081 | 0.6867 |
| ALASSO | 0.9929 | 0.0383 | 0.7458 | 0.7736 | 0.7636 |
| Normal Part | | | | | |
| LASSO | 1.3307 | 0.0144 | 0.9978 | 0.4887 | 0.6516 |
| ALASSO | 1.1517 | 0.0110 | 0.9878 | 0.7794 | 0.8461 |
| Two Parts | | | | | |
| LASSO | 2.3758 | 0.0267 | 0.9071 | 0.5466 | 0.6692 |
| ALASSO | 2.1446 | 0.0246 | 0.8596 | 0.7766 | 0.8048 |
| 2. When $n = 300$ | | | | | |
| Binomial Part | | | | | |
| LASSO | 0.9957 | 0.0197 | 0.9182 | 0.5413 | 0.6770 |
| ALASSO | 0.9660 | 0.0178 | 0.8478 | 0.7823 | 0.8058 |
| Normal Part | | | | | |
| LASSO | 1.1483 | 0.0059 | 1.0000 | 0.4958 | 0.6571 |
| ALASSO | 1.0677 | 0.0044 | 0.9995 | 0.8042 | 0.8667 |
| Two Parts | | | | | |
| LASSO | 2.1440 | 0.0128 | 0.9567 | 0.5178 | 0.6670 |
| ALASSO | 2.0337 | 0.0111 | 0.9192 | 0.7936 | 0.8363 |

**Table 6. The simulation results of the setting $q = 25, \rho = 0.6$.**

| Method | MPSE | MSE | Sensitivity | Specificity | Accuracy |
|---|---|---|---|---|---|
| 1. When $n = 150$ | | | | | |
| Binomial Part | | | | | |
| LASSO | 0.9371 | 0.0519 | 0.7453 | 0.6865 | 0.7077 |
| ALASSO | 0.8845 | 0.0742 | 0.6400 | 0.7655 | 0.7203 |
| Normal Part | | | | | |
| LASSO | 1.3113 | 0.0240 | 0.9530 | 0.5368 | 0.6700 |
| ALASSO | 1.1470 | 0.0223 | 0.9475 | 0.7519 | 0.8145 |
| Two Parts | | | | | |
| LASSO | 2.2484 | 0.0379 | 0.8431 | 0.6094 | 0.6888 |
| ALASSO | 2.0315 | 0.0483 | 0.7849 | 0.7585 | 0.7674 |
| 2. When $n = 300$ | | | | | |
| Binomial Part | | | | | |
| LASSO | 0.8882 | 0.0305 | 0.7318 | 0.6258 | 0.6999 |
| ALASSO | 0.8606 | 0.0334 | 0.7604 | 0.7776 | 0.7714 |
| Normal Part | | | | | |
| LASSO | 1.1400 | 0.0103 | 0.9895 | 0.4976 | 0.6550 |
| ALASSO | 1.0605 | 0.0084 | 0.9840 | 0.7869 | 0.8500 |
| Two Parts | | | | | |
| LASSO | 2.0204 | 0.0204 | 0.8060 | 0.5598 | 0.6775 |
| ALASSO | 1.9211 | 0.0209 | 0.8656 | 0.7824 | 0.8107 |

- In terms of the results of Specificity, the values of two methods in the binary part are better than the continuous part. As the correlation coefficient $\rho$ increases from 0 to 0.6, the Specificity of Lasso method improves, but the Specificity of adaptive Lasso method has no significant change. In a word, the adaptive Lasso method shows better results in different settings.

- From the overall Accuracy results, the variable selection effect based on the adaptive Lasso method is better than Lasso method. And with the increase of $n$, the Accuracy of the two methods is getting higher.

## 5 Case application

In medicine, investigators often encounter semicontinuous data. For example, in studies of dietary intake, the dietary intake data often contain a large number of zeros, because some food components are only occasionally consumed by patients. In this section, the proposed methods are applied to the CHEF data to analyze the factors influencing the dietary intake of patients.

CHEF (Cultivating Healthy Environments in Families with Type 1 Diabetes) is an 18-month randomized trial to evaluate the efficacy of a family-based behavioral intervention that integrated motivational interviewing, active learning, and applied problem-solving to increase intake of whole plant foods among youth with type 1 diabetes. In the CHEF study, a total of 136 children with type 1 diabetes participated, and CHEF data are obtained by collecting dietary data over six periods, including the baseline period (before the intervention) and the intervention period, based on the diet records of the patients. A total of 12 food components are recorded in CHEF data, among which 8 food are continuous variables because they are consumed by patients every day. The other 4 foods Total Fruit (TF), Whole Fruit (WF), Dark Green/Orange Vegetables & Legumes (DOL) and Whole Grain (WG) have excessive zeros, which due to the patient's occasional intake of these food components. Therefore, CHEF data contains four semicontinuous variables which is a typical semicontinuous data. In addition, 28 candidate variables that may affect patient intake are collected from the CHEF trial. In pursuit of precision intervention, investigators are interested in the factors that influence the dietary intake of children with type 1 diabetes before behavioral interventions. Details of the study design, randomization procedures and treatment conditions can be found in Nansel et al. (2015) [23].

The variable selection methods proposed are mainly aimed at the BNT model. In order to avoid the wrong conclusion caused by the poor fitting of the model, we first carry out Shapiro-Wilks normality test for the continuous part of the four semicontinuous variables. The specific test results are shown in Table 7. And it appears that only the continuous part of DOL response variable follows a normal distribution. Therefore, we build a BNT regression model for this variable, and use the proposed methods based on Lasso and adaptive Lasso penalty function to select the key factors. The estimated coefficients and selected factors by the two methods are respectively showed in Tables 8 and 9. From the results, we can conclude as follows. For the binary part and the continuous part, different subsets of variables are selected based on different methods. Some variables are both selected by the two methods. Although the estimated coefficient values by the two methods are different, the positive or negative signs of their estimated coefficient are the same. For example, the binary part of C_QOLEMO (Child Generic quality of life, Emotional subscale) and C_QOLSCH (Child Generic quality of life, School subscale) variables. Both methods can make variable selection while avoiding overfitting.

**Table 7. The Shapiro-Wilks test results of four semicontinuous response variables.**

| Variables | TF | WF | DOL | WG |
|---|---|---|---|---|
| p-value | 0.0003 | 0.0409 | 0.5077 | 0.0324 |

**Table 8. The estimation results of Lasso penalty method.**

| Variables | Binomial Part | Normal Part | Variables | Binomial Part | Normal Part |
|---|---|---|---|---|---|
| Intercept | -1.4899 | -1.3865 | P_QOLSOC | | |
| HEI2005 | 0.0423 | 0.0036 | C_QOLG | | |
| WPF | | 0.2828 | C_QOLEMO | -0.0346 | |
| P_HEB | | | C_QOLPSY | | |
| P_HES | | | C_QOLSCH | -0.1518 | |
| P_HEO_P | 0.0044 | | C_QOLSOC | | |
| P_HEO_N | | | C_HEB | | |
| P_TSR_AUT | | | C_HEO_P | | |
| P_TSR_CON | | | C_HEO_N | | |
| P_TSR_AM | | | C_HES | | |
| P_NKS_PCT | | | C_TSR_AUT | | |
| P_QOLG | | | C_TSR_CON | | |
| P_NKS_QOLEMO | | | C_TSR_AM | | |
| P_NKS_QOLPSY | | | C_NKS_PCT | 0.0072 | |
| P_NKS_QOLSCH | | | | | |

**Table 9. The estimation results of adaptive Lasso penalty method.**

| Variables | Binomial Part | Normal Part | Variables | Binomial Part | Normal Part |
|---|---|---|---|---|---|
| Intercept | -1.0058 | -1.1476 | P_QOLSOC | | |
| HEI2005 | | | C_QOLG | | |
| WPF | | 0.3348 | C_QOLEMO | -0.2374 | |
| P_HEB | | | C_QOLPSY | | -0.3890 |
| P_HES | | | C_QOLSCH | -0.2774 | |
| P_HEO_P | | | C_QOLSOC | | |
| P_HEO_N | | | C_HEB | | -0.1924 |
| P_TSR_AUT | | | C_HEO_P | | |
| P_TSR_CON | | | C_HEO_N | | |
| P_TSR_AM | | | C_HES | | |
| P_NKS_PCT | | | C_TSR_AUT | | |
| P_QOLG | 0.4099 | 0.4715 | C_TSR_CON | | |
| P_NKS_QOLEMO | | | C_TSR_AM | | |
| P_NKS_QOLPSY | | | C_NKS_PCT | | |
| P_NKS_QOLSCH | | | | | |

In addition, we use the -2Loglik, AIC and BIC criteria to compare the variable selection effects of the two methods. In order to further verify the effectiveness of the proposed method on real data, the *K*-fold cross-validation method is adopted. The specific results are shown in Table 10. It shows that the adaptive Lasso method respectively has smaller -2Loglik, AIC, BIC and ME values for the whole model (Two Parts). Moreover, for the binary part and the continuous part, the adaptive Lasso method also has smaller -2Loglik, AIC, BIC and ME values. Therefore, the results show that the proposed method based on the adaptive Lasso penalty function has better fitting effect. The results of ME show that the adaptive Lasso method has a small error for real data. That is, the variables selected by the adaptive Lasso method are a more efficient subset of important variables.

Based on the results of modeling, we discuss the factors influencing the intake of DOL from three aspects.

- Factors in the binary part, that is, explanatory variables that affect whether patients consume DOL foods, such as P_QOLG (Parent Generic quality of life, total score),

**Table 10. The -2LogLik, AIC, BIC and ME values of the two methods.**

| Method | Lasso | | | Adaptive Lasso | | |
|---|---|---|---|---|---|---|
| | Binomial Part | Normal Part | Two parts | Binomial Part | Normal Part | Two parts |
| -2LogLik | 116.2524 | 253.7255 | 369.9779 | 113.9831 | 242.7999 | 356.7830 |
| AIC | 126.5024 | 257.7255 | 384.2279 | 119.9831 | 250.7999 | 370.7830 |
| BIC | 141.1257 | 262.7251 | 404.3665 | 128.721 | 260.79991 | 391.1716 |
| ME | 1.2634 | 1.0378 | 2.3012 | 1.2130 | 0.9585 | 2.1715 |

C_QOLEMO, C_QOLSCH. The factors with a positive coefficient would increase the likelihood of ingestion of DOL, such as P_QOLG. The factors with a negative coefficient would reduce the likelihood of patients ingesting of DOL, such as C_QOLEMO, C_QOLSCH.

- Factors of the continuous part, that is, the interpretative variables that affect the amount of DOL ingested by patients, such as WPF, P_QOLG, C_QOLPSY (Child Generic quality of life, Psychosocial subscale), C_HEB (Child Healthy Eating Barriers). The factors with a positive coefficient would increase the patient's intake of DOL, such as WPF, P_QOLG. The factors with a negative coefficient would reduce the patient's intake of DOL, such as C_QOLPSY, C_HEB.
- The factors with a coefficient of 0, indicating no effect on DOL food intake, such as P_HEB (Parent Healthy Eating Barriers), P_HES (Parent Self-Efficacy). Therefore, the adaptive Lasso penalized BNT regression model can be used to further analyze the important factors affecting the dietary intake of patients.

# 6 Concluding remarks

In this paper, we mainly discuss the variable selection problem of BNT regression model. In the framework of the penalized likelihood function method, we respectively propose the variable selection method based on Lasso or adaptive Lasso penalty function. Simulation results show that both methods can select variables for the BNT regression model, but the performance of the adaptive Lasso method is better than the Lasso method. In addition, we exemplified the proposed methods with a real dietary data to analyze factors influencing the dietary intake of patients, and the results illustrate the advantages of the proposed method.

With the advances of information technology, semicontinuous data occur in more and more fields. The proposed methods can be applied to other research fields. For example, in insurance fields, the cumulative loss amount data often contains a large number of zeros. At this point, the semicontinuous two-part regression model can be used to analyze the influencing factors of cumulative loss, and the proposed methods can select the more important risk factors. Although this paper has achieved some breakthrough results, there are still some research problems that need to be further addressed. For instance, as more effective penalty functions are proposed, the variable selection method of semicontinuous two-part regression models based on these penalty functions also deserves further investigation. Considering the distribution characteristics of the continuous part of the semicontinuous data, different two-part models can be constructed, such as the Bernoulli-Gamma model and Bernoulli-Skewed Normal model. The variable selection problem of these regression models is one of the subsequent research problems. In addition, our research is mainly for the case of $p \ll n$, when $p \gg n$, the variable selection method of semicontinuous two-part regression model is also one of the contents of our current research.

## Supporting information

**S1 Appendix. In the Appendix, we first provide more additional simulation studies to further emphasize the performance of the proposed methods, and then we share the code in a way that follows best practices and promotes repeatability and reuse.**
(DOCX)

## Acknowledgments

The authors were grateful to the referees, associate editor, and the editor for their valuable comments and suggestions. The authors thanked Dr. Tonja Nansel for helpful discussions on the CHEF study.

## Author contributions

**Data curation:** Yahui Lu.

**Investigation:** Tao Jiang.

**Methodology:** Aiyi Liu, Tao Jiang.

**Validation:** Yahui Lu.

**Writing – original draft:** Yahui Lu.

**Writing – review & editing:** Yahui Lu.

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
