## [Decision Letter · Decision Letter 0]

PONE-D-24-37521The Variable Selection of Two-part Regression Model for Semicontinuous DataPLOS ONE

Dear Dr. Yahui,

Thank you for submitting your manuscript to PLOS ONE. After careful consideration, we feel that it has merit but does not fully meet PLOS ONE’s publication criteria as it currently stands. Therefore, we invite you to submit a revised version of the manuscript that addresses the points raised during the review process.

We look forward to receiving your revised manuscript.

Kind regards,

Flavio A. Ziegelmann, Ph.D.

Academic Editor

PLOS ONE

3. Thank you for stating the following financial disclosure: [Research of A. Liu is supported by the

Intramural Research Program of the Eunice Kennedy Shriver National Institute of Child Health and Human Development. Lu’s work is supported by the research foundation of Zhejiang University of Science and Technology (2023QN094) and Hangzhou philosophy and social science planning project (Z23JC042).]. Please state what role the funders took in the study. If the funders had no role, please state: "The funders had no role in study design, data collection and analysis, decision to publish, or preparation of the manuscript." If this statement is not correct you must amend it as needed. Please include this amended Role of Funder statement in your cover letter; we will change the online submission form on your behalf.

Additional Editor Comments:

The two reviewers had very different feelings about the paper contribution. Reading the paper myself, I have to say that my impression is closer to that of reviewer 2, which recommends its rejection due to the absence of novel statistical methods and to limited simulation and empirical exercises.

Nevertheless, even though the methodological contribution is very minor at most, the paper is well written and deals with an interesting problem.

So, also considering reviewer 1 evaluation, my recommendation is for a major revision, in case the authors are motivated to do so. It should incorporate changes and additions according to all the points raised by the reviewers, especially reviewer 2. There are some deep points raised there, which should all be dealt with to build a newer version of the paper.

Reviewers' comments:

Reviewer's Responses to Questions

**Comments to the Author**

1. Is the manuscript technically sound, and do the data support the conclusions?

Reviewer #1: Yes

Reviewer #2: Partly

2. Has the statistical analysis been performed appropriately and rigorously? 

Reviewer #1: Yes

Reviewer #2: No

3. Have the authors made all data underlying the findings in their manuscript fully available?

Reviewer #1: No

Reviewer #2: Yes

4. Is the manuscript presented in an intelligible fashion and written in standard English?

Reviewer #1: Yes

Reviewer #2: Yes

5. Review Comments to the Author

Reviewer #1: The data was not made available. Also, I suggest that the authors specify which software was used to conduct the analyses. Additionally, if feasible, making the code for the simulations and applications available to readers would enhance the paper’s transparency and reproducibility.

Reviewer #2: In this paper, authors proposed methods for variable selection for the two-part regression model specifically, the Bernoulli-Normal two-part (BNT) regression model. The variable selection method is based on adaptive Lasso penalty function. The following are detailed comments:

1. The usage of normal model for the non-zero part is limited. In reality, often the continuous measure does not follow normal distribution.

2. It would be interesting to see the model performance for the following scenarios

a) Varying Zero Proportions including low, moderate, and high zero proportions

b) Different Variable-Effect Strengths including strong and weak signal between predictors and outcome in one or both parts of the model.

c) Different Correlations Between Parts. The current simulation assumes independent parts. However, it will be useful to test the robustness of the model under the violation of the independence assumption.

d) It is not clear the model performance with high-dimensional data. The current simulation assumes the largest number of covariates in the model is 25.

e) Simulate scenarios where some predictors only influence the zero part, while others only influence the non-zero part. This will test the method’s capacity to select variables that are important for only one part of the model. The current simulation assumes the same covariates for both parts of the model.

f) Simulate cases where the model is misspecified for one or both parts, to test how robust the variable selection method is to model misspecification.

3. There is no software available publicly. It is hard for other researchers to implement the proposed approach.

6. PLOS authors have the option to publish the peer review history of their article (what does this mean?). If published, this will include your full peer review and any attached files.

Reviewer #1: No

Reviewer #2: No

---

## [Author Response · Author response to Decision Letter 1]

16 Feb 2025

Dear Editors and Reviewers,

We sincerely thank you for your careful reading of our manuscript and for providing constructive comments. We have carefully taken into consideration the comments from you. Accordingly, we have made substantial changes based on these comments and provided detailed point-by-point responses.

We have responded to your specific concerns point by point, please see the separate file labeled "Response to Reviewers". For your convenience, changes made accordingly are highlighted in the revised manuscript.

---

## [Decision Letter · Decision Letter 1]

The Variable Selection of Two-part Regression Model for Semicontinuous Data

PONE-D-24-37521R1

Dear Dr. Yahui,

We’re pleased to inform you that your manuscript has been judged scientifically suitable for publication and will be formally accepted for publication once it meets all outstanding technical requirements.

Kind regards,

Flavio A. Ziegelmann, Ph.D.

Academic Editor

PLOS ONE

Additional Editor Comments:

I agree with the two reviewers. The authors have satisfactorily dealt with all the issues raised by them in the first round. The only pending issue is the possible lack of data availability, which is still mentioned by one of the reviewers.

**Comments to the Author**

Reviewer #1: All comments have been addressed

Reviewer #2: All comments have been addressed

2. Is the manuscript technically sound, and do the data support the conclusions?

Reviewer #1: Yes

Reviewer #2: Yes

3. Has the statistical analysis been performed appropriately and rigorously? 

Reviewer #1: Yes

Reviewer #2: Yes

4. Have the authors made all data underlying the findings in their manuscript fully available?

Reviewer #1: Yes

Reviewer #2: No

5. Is the manuscript presented in an intelligible fashion and written in standard English?

Reviewer #1: Yes

Reviewer #2: Yes

---

## [Editor Report · Acceptance letter]

PONE-D-24-37521R1

PLOS ONE

Dear Dr. Yahui,

I'm pleased to inform you that your manuscript has been deemed suitable for publication in PLOS ONE. Congratulations! Your manuscript is now being handed over to our production team.

Kind regards,

on behalf of

Professor Flavio A. Ziegelmann

Academic Editor

PLOS ONE